# Flavonoid Baicalein Suppresses Oral Biofilms and Protects Enamel Hardness to Combat Dental Caries

**DOI:** 10.3390/ijms231810593

**Published:** 2022-09-13

**Authors:** Hong Chen, Sihong Xie, Jing Gao, Liwen He, Wenping Luo, Yunhao Tang, Michael D. Weir, Thomas W. Oates, Hockin H. K. Xu, Deqin Yang

**Affiliations:** 1Department of Endodontics, Stomatological Hospital of Chongqing Medical University, Chongqing 404100, China; 2Chongqing Key Laboratory of Oral Diseases and Biomedical Sciences, Stomatological Hospital of Chongqing Medical University, Chongqing 404100, China; 3Chongqing Municipal Key Laboratory of Oral Biomedical Engineering of Higher Education, Chongqing 404100, China; 4Chongqing Key Laboratory of Oral Diseases and Biomedical Sciences, 426 Songshi North Road, Yubei District, Chongqing 401147, China; 5Department of Advanced Oral Sciences and Therapeutics, University of Maryland Dental School, Baltimore, MD 21201, USA; 6Center for Stem Cell Biology and Regenerative Medicine, University of Maryland School of Medicine, Baltimore, MD 21201, USA; 7Marlene and Stewart Greenebaum Cancer Center, University of Maryland School of Medicine, Baltimore, MD 21201, USA

**Keywords:** baicalein, *Streptococcus mutans*, *Candida albicans*, biofilm inhibition, dental caries

## Abstract

The objectives of this study were to investigate the effects of a novel method using flavonoids to inhibit *Streptococcus mutans* (*S. mutans*), *Candida albicans* (*C. albicans*) and dual-species biofilms and to protect enamel hardness in a biofilm-based caries model for the first time. Several flavonoids, including baicalein, naringenin and catechin, were tested. Gold-standard chlorhexidine (CHX) and untreated (UC) groups served as controls. Optimal concentrations were determined by cytotoxicity assay. Biofilm MTT, colony-forming-units (CFUs), biofilm biomass, lactic acid and polysaccharide production were evaluated. Real-time-polymerase-chain reaction (qRT-PCR) was used to determine gene expressions in biofilms. Demineralization of human enamel was induced via *S. mutans*-*C. albicans* biofilms, and enamel hardness was measured. Compared to CHX and UC groups, the baicalein group achieved the greatest reduction in *S. mutans*, *C. albicans* and *S. mutans*-*C. albicans* biofilms, yielding the least metabolic activity, polysaccharide synthesis and lactic acid production (*p* < 0.05). The biofilm CFU was decreased in baicalein group by 5 logs, 4 logs, 5 logs, for *S. mutans*, *C. albicans* and *S. mutans*-*C. albicans* biofilms, respectively, compared to UC group. When tested in a *S. mutans*-*C. albicans* in vitro caries model, the baicalein group substantially reduced enamel demineralization under biofilms, yielding an enamel hardness that was 2.75 times greater than that of UC group. Hence, the novel baicalein method is promising to inhibit dental caries by reducing biofilm formation and protecting enamel hardness.

## 1. Introduction

Dental caries is one of the most widespread and costly biofilm-mediated oral infectious diseases, affecting people of all ages worldwide [1]. Among the several hundred bacterial species in the dental plaque, *Streptococcus mutans* (*S. mutans*) is a principal causative agent of caries [2]. *Candida albicans* (*C. albicans*) is one of the most common fungi in mouth [3,4], and oral colonization of *C. albicans* is increased in root caries (RC) and early childhood caries (ECC) lesions [5,6]. *C. albicans* has an extraordinary acidogenic capacity and acid tolerance, and its association with *S. mutans* results in increased exopolysaccharides (EPS) formation and acid production and yield-enhancing biofilm cariogenicity [7,8]. Indeed, the positive correlation between *C. albicans* and the risk of dental caries has been demonstrated [9], and individuals with higher yeast counts in saliva and dental biofilms have a higher incidence of RC and ECC [6,7,10].

Preventing biofilm formation is a key to avoid the occurrence of dental caries [11]. Chlorhexidine (CHX) is a potent dental agent against oral infections, due to its broad-spectrum antimicrobial efficacy [12,13]. CHX can increase the cellular membrane permeability of bacteria or fungi by interacting with the anionic receptors on the cell surface [14,15]. CHX is widely used in clinical dentistry, and its therapeutic benefit has been demonstrated in reducing dental plaque [16]. However, bacterial spores and mycobacteria are highly resistant to CHX [17]. Thus, novel and effective methods should be explored to fight and prevent cariogenic biofilms.

Flavonoids are a family of polyphenolic compounds possessing great antibacterial and antifungal action [18]. Studies have shown that flavonoids have potentially beneficial effects as antimicrobial agents in the therapy for human disease [19,20]. In view of the fact that flavonoids represent an emerging threat of dental microbe, the present study was designed to evaluate the effect of selected single flavonoids (baicalein, naringenin, and catechin) for their activities against dental biofilm bacteria and fungi. In addition, to date, there were few studies on the role of flavonoids in affecting *S. mutans*, *C. albicans* and the dual-species biofilm formation [21,22].

Therefore, the objectives of this study were to investigate, for the first time, that: (1) flavonoids could inhibit single (*S. mutans* or *C. albicans*) and dual-species (*S. mutans*-*C. albicans*) cariogenic biofilm formation, with CHX as control; (2) flavonoids would substantially reduce enamel demineralization and increase the enamel hardness under *S. mutans*-*C.*
*albicans* biofilm.

## 2. Results

The concentration of antimicrobial agent in this experiment was screened using a cytotoxicity experiment. As shown by the results of CCK-8 assay (Figure 1), the absorbance rates decreased with increasing concentrations based on the negative values observed in the negative control group. As gold standard, CHX were chosen as our positive control to evaluate the cell toxicity. Based on the cell cytotoxicity test, the antibacterial concentrations were selected for our study as follows: 0.250 mg/mL of baicalein, 1.000 mg/mL of naringenin, 0.250 mg/mL of catechin, and 0.00031 mg/mL of CHX, respectively.

Representative SEM micrographs of typical biofilms on adhesive disks in different groups are shown in Figure 2. The five groups (CHX, baicalein, naringenin, catechin, and untreated control (UC)) were labeled in the images. The two types (*S. mutans* and dual-species) were labeled in these groups. Compared to the UC groups, biofilms of CHX, baicalein and naringenin groups on the adhesive disks did not fully develop even after 4 h. While the 0.250 mg/mL baicalein groups had the greatest antibacterial activity, only a few bacteria or fungi were observed on the surface of the disk.

The CV staining image is presented in Figure 3A, and further biofilm biomass results are shown in Figure 3B (mean ± sd; *n* = 6). The drugs greatly reduced the biomass significantly containing flavonoid or CHX, compared to the UC group (*p* < 0.01), and the inhibition effect of baicalein is the greatest and much better than that of CHX (*p* < 0.01).

Six standard glucose concentrations of 0, 10, 20, 30, 40 and 50 μg/mL were used to plot the standard curve of OD_620nm_ versus polysaccharide concentrations. A linear curve y = 0.0486x + 0.0614 was obtained, and the coefficient of determination (R^2^) was 0.9829. Figure 3C presents the polysaccharide synthesis with or without drugs (mean ± sd; *n* = 6). Two main results were obtained here (Figure 3C): (1) flavonoids and CHX remarkably decreased the polysaccharide production, compared to UC group (mean ± sd; *n* = 6; *p* < 0.01). (2) Using baicalein, the polysaccharide synthesis decreased more than 90%, which is much better than CHX control (*p* < 0.01).

Figure 3D plotted that *S. mutans* and dual-species biofilms had relatively higher levels of lactic acid production compared to *C. albicans* biofilms. The lactic acid production of single- and dual-species biofilms was greatly inhibited on the flavonoids and CHX groups (*p* < 0.01), and lactic acid production of baicalein group on biofilms was the lowest in all groups.

The antibacterial effects of flavonoids and CHX against the *S. mutans*, *C. albicans* and *S. mutans*-*C. albicans* biofilms are shown in Figure 4 (mean ± sd; *n* = 6). MTT metabolic activity and colony forming unit (CFU) were markedly decreased in the baicalein group (*p* < 0.01). Compared with the UC group, the CFU of *S. mutans* biofilm was decreased by nearly 5 logs and 4 logs in *S. mutans* and *S. mutans*-*C. albicans* biofilms, respectively, at 0.250 mg/mL baicalein (Figure 4B). While 4 logs and 1 log in *C. albicans* and *S. mutans*-*C. albicans* biofilms were reduced for *C. albicans* CFU at 0.250 mg/mL baicalein (Figure 4C). The use of baicalein caused the greatest reduction in biofilm activity, while the CHX group appeared to have no effect.

CLSM showed that only small clusters of bacterial cells were observed in the baicalein and naringenin groups, while the CHX and catechin groups were similar with UC group (Figure 5A). It was concluded that baicalein and naringenin impaired EPS production and biofilm formation. Furthermore, we calculated the EPS/bacterial volume ratio, and baicalein exhibited the least value, which indicated the best anti-bacterial effect of baicalein in the EPS architecture development (Figure 5B,C). In addition, there was almost no EPS production of *C. albicans* biofilms in all groups, and therefore, we have deleted the results for *C. albicans* biofilms in Figure 5C.

The expression profiles of *gtfB*/*C*/*D* and *comC*/*D*/*E* genes are shown in Figure 6. After the biofilm cultured with flavonoids or CHX, the transcription levels of *gtfB*/*C*/*D* and *comC*/*D*/*E* genes statistically declined (*p* < 0.05). Compared with the UC group in the *S. mutans* biofilm, the expression of *gtfB*/*C*/*D* in the baicalein group was down-regulated to 0.58, 0.56 and 0.35 folds, respectively (Figure 6A), while in the naringenin group, it was down-regulated to about 0.20 folds significantly. In dual-species biofilm, catechin down-regulate the expression of *gtfB*/*C*/*D* gene to 0.69–0.73 folds. In addition, the data revealed that the transcriptional levels of *comC/D*/*E* were significantly decreased in dual-species biofilm (Figure 6B). The expression of *comC*/*D*/*E* in the baicalein group was down-regulated to 0.34–0.70 folds, while in the naringenin group, it was down-regulated to 0.25–0.40 folds (Figure 6B).

Enamel hardness at the enamel surface is plotted in Figure 7 (mean ± sd; *n* = 6). The sound enamel hardness was 3.08 ± 0.21 GPa. After 14 days under the *S. mutans-C. albicans* biofilm acid attack, enamel hardness decreased in every group, but the baicalein group had the highest enamel hardness (*p* < 0.01). These results showed that: (1) Flavonoids baicalein substantially increased the enamel hardness, by 2.75 folds, compared to the UC group. (2) The efficacy of baicalein and naringenin was much better than CHX in protecting the enamel hardness.

## 3. Discussion

Flavonoids are a promising therapy against dental caries due to their antimicrobial activity and low toxicity. In the present study we investigated flavonoids with regard to their cytotoxicity on oral HOK cells and antimicrobial effects on cariogenic biofilm, using CHX as control group. There existed a certain usage concentration with regard to their cytotoxic activities in vitro. The antimicrobial ability observation and biofilm-based caries model proved that baicalein exerted instant plaque-inhibiting effects, helping control biofilm formation and protect tooth structure, which suggested that baicalein exerted great clinical application potentials.

Biocompatibility is a necessary indicators for evaluating all kinds of biomedical agents [23]. The toxicity to HOK cells was tested to explore the potential of flavonoids for dental clinical application [24]. An ideal antimicrobial agent should have effectiveness killing microbes while being safe to the host cell [23,25]. Therefore, in vitro and in vivo potential toxicity of antimicrobial agents should be evaluated before clinical use. However, sometimes in vitro studies used to evaluate drug toxicity tend to display cell toxicity, but they are still widely used in clinical treatment under certain conditions [23,26]. For example, 0.2% CHX is widely used in mouthwash, but 0.2% CHX in the present study revealed obvious toxic effects on human HOK cells viability. In the present study, we also evaluated the cytotoxic activities of flavonoids. Following current studies [23,27], we obtained the drug concentrations of more than 85% cell viability, and the results indicated that flavonoids concentrations of 0.250 mg/mL in baicalein, 1.000 mg/mL in naringenin and 0.250 mg/mL in catechin have good biocompatibility with HOK cells. Therefore, the concentrations above were used in the following experiments.

*C. albicans* is a Gram-positive fungal microorganism, which plays an important role in the development of dental caries because of its positive interactions with *S. mutans* [8]. The interaction between *C. albicans* and *S. mutans* can promote the occurrence and development of root caries and ECC [5,6]. *C. albicans* can enhance the sugar metabolism pathways of *S. mutans* [8]. *S. mutans* possesses an exceptional ability to produce EPS, and the EPS matrix is a key virulence factor for cariogenic biofilm [28,29]. As nutrients and the core of the matrix scaffold, EPS can enhance biofilm accumulation and stability [28,29]. In the present study, flavonoids disrupted the polysaccharide synthesis and biofilm formation in the *S. mutans*, *C. albicans* and the dual-species biofilms. Therefore, the use of drugs flavonoids might be a potential strategy for caries prevention since previous reports have indicated that the EPS initiates the adhesion for first colonizers and is critical for dental biofilms development.

In the present study, baicalein flavonoids substantially reduced the biofilm metabolic activity by more than 4 orders of magnitude. In sharp contrast, the biofilm CFU reduction in CHX had no significance compared to the UC group. The *S. mutans* and *C. albicans* in biofilms may be resistant to CHX [11]. Therefore, baicalein possesses anti-microbe effects of flavonoids that are much better than CHX.

Acidogenicity in *S. mutans* is a virulence factor associated with cariogenicity [1,30]. Biofilms can produce organic acids (mainly lactic acid) and induce teeth demineralization [28,30]. In the present study, the lactic acid secretion of biofilms decreased significantly in all drug groups, especially the baicalein group. The usage of baicalein achieved the greatest reduction in lactic acid production, more than 90% in *S. mutans* and *dual-species* biofilms. Such a major drop in biofilm acid production is expected to effectively inhibit dental caries, which warrants further study.

Sucrose-dependent adhesion is one of the major virulence factors [31,32]. The adhesion process is mainly mediated by glucans, which are synthesized from sucrose by glucosyltransferase (GTF) enzymes [33]. GTFB, GTFC and GTFD are the three important GTF enzymes [34]. The corresponding genes involved are *gtfB*, *gtfC*, and *gtfD*. The mRNA expression of *gtfB*, *gtfC* and *gtfD* of *S. mutans* and dual-species biofilm was examined using qRT-PCR in this study. The results showed that naringenin and baicalein could inhibit the mRNA expression for *gtfB*/*C*/*D* in *S. mutans* biofilm. The relative expressions of *gtfB*/*D* for *S. mutans*-*C. albicans* biofilms culturing in baicalein, of *gtfB*/*C* in naringenin and of *gtfB*/*C*/*D* in catechin also decreased. This demonstrated that flavonoids could reduce the sucrose-dependent adhesion of *S. mutans* and further decrease single- and dual-biofilm formation, which may neutralize the increase of sucrose independent adhesion, finally showing an inhibition effect.

The QS system controls biofilm formation and virulence factors release in cariogenic biofilms [35]. This system plays a key role in the competition and coexistence between microbe in biofilms and other significant processes [36]. In *S. mutans*, the most common intraspecific QS system is the competence-stimulating peptide (CSP)-QS system [37]. Hence, *comC*/*D*/*E* genes were tested, and the results proved that these genes were downregulated after incubation with naringenin and baicalein in the present study. In other words, naringenin and baicalein may inhibit the synthesis of QS system, which would reduce the internal signal transduction in *S. mutans* or *S. mutans* and *C. albicans*.

In the *S. mutans*-*C. albicans* in vitro caries model, the protection effects of flavonoids were verified. The hardness of the enamel in drug groups was all higher than that of UC groups. Compared to the CHX and UC groups, the enamel hardness in baicalein groups showed a significant increase at the enamel surface under 14 days of biofilm acid attack. These results demonstrate that baicalein could inhibit biofilm formation and hindered cariogenic activities, effectively inhibiting tooth demineralization and protecting tooth structure.

When the single- and dual-species biofilms were cultured in the flavonoids, the biofilm formation, metabolic activities and cariogenic activities were all significantly reduced. In addition, we proved the effect of flavonoids baicalein on inhibiting tooth demineralization under *S. mutans*-*C. albicans* biofilms for longer than 14 days. This study demonstrates the excellent antibacterial and antifungal effects of baicalein. Based on these results, the usage of baicalein could successfully inhibit dental caries pathogen biofilm formation. Further studies are needed to investigate the effect of baicalein in vivo, and more species biofilms also need to be studied regarding the complexity of oral microbe.

Regarding potential clinical applications, there is a major need in dentistry to decrease biofilm formation, as well as to inhibit caries and protect tooth structures. The present study demonstrated for the first time that the use of topical baicalein-containing solutions effectively suppressed caries associated *S. mutans* or *C. albicans*, yielding much greater enamel hardness under biofilm attacks. Dental applications of this novel method could include baicalein-based drugs for anti-caries clinical applications, such as dentifrices, mouthrinses, moisturizing gels, varnishes and chewing gums. Further studies are needed to investigate and realize these potentials.

## 4. Materials and Methods

### 4.1. Chemicals, Bacterial and Fungal Strains and Growth Conditions

Baicalein, naringenin, catechin and CHX were purchased commercially (Solarbio, Beijing, China). The use of all the bacterial species was approved by the Institutional Ethics Committee of Stomatological Hospital of Chongqing Medical University. *S. mutans* strain UA159 (ATCC 700610) and *C. albicans* SC5314 (ATCC MYA-2876) were provided by Sichuan University [3,15]. Precultures of *S. mutans* were grown in brain–heart infusion (BHI) medium at 37 °C under 5% CO_2_ [11]. Precultures of *C. albicans* were grown in YPD medium containing 1% yeast extract, 2% peptone, and 2% D-glucose at 37 °C under 5% CO_2_ as well [3]. The *S. mutans* biofilms were cultured in BHI supplemented with 1% sucrose (*wt*/*vol*) (BHIS). As for *C. albicans* and dual-species biofilms, YNBB (0.67% YNB, 75 mM Na_2_HPO_4_-NaH_2_PO_4_, 2.5 mM N-acetylglucosamine, 0.2% casamino acids, and 0.5% sucrose) was used [3].

### 4.2. Biofilm Formation

Precultures of *S. mutans* and *C. albicans* from single colonies were incubated overnight [3,11]. Then *S. mutans* were diluted to 2 × 10^6^ CFU/mL into fresh BHIS, and *C. albicans* were diluted to 2 × 10^4^ cell/mL into YNBB medium, with or without drugs. A volume of 2 mL bacteria dilution was cultured to form biofilms in 24-well plates. For dual-species biofilm, inoculum for the experiment was adjusted to 2 × 10^6^ cell/mL of *S. mutans*, and 2 × 10^4^ cell/mL of *C. albicans* [3]. Equal volumes of each strain (200 uL) and 1.6 mL YNBB medium with and without drugs were also incubated in 24-well plates for dual-species biofilm formation [38]. The plates were incubated at 37 °C under 5% CO_2_ for 4 h. The 4 h biofilms were then used for subsequent experiments.

### 4.3. Cytotoxicity Assays

Human oral keratinocyte cells (HOK cells) were used for cytotoxicity assays [39]. The HOK cells were cultured in a low-glucose Dulbecco’s modified eagle’s medium (DMEM, Gibco, Grand Island, NY, USA) supplemented with 10% fetal bovine serum (FBS) and 1% penicillin-streptomycin (Invitrogen, Carlsbad, CA, USA) (control media) [39,40]. The cells were incubated at 37 °C with 5% CO_2_, and the culture medium was changed every 2–3 days [40]. Each group, with or without drugs, and 8000 HOK cells were placed into a well of 96-well plates. Control medium was added accordingly. A cell counting kit (CCK-8, Dojindo, Tokyo, Japan) was used to evaluate cell proliferation [40]. Three replicates in each group were used for this assay (*n* = 3). After 24 h, the 96-well plate was washed twice with PBS. A volume of 100 uL of growth medium with 10% CCK-8 for 2 h was added into each well, and the cell proliferative rate was determined via measuring the absorbance at OD_450nm_ using a microplate reader (SpectraMax M5, Molecular Devices, San Jose, CA, USA) [40]. Optimal tested concentrations were the drug concentrations with cell viability of above 85% [23]. Then the drugs (flavonoids and CHX) and UC groups were used in the following experiments.

### 4.4. Scanning Electron Microscopy (SEM)

The biofilm inhibition effects were observed using a scanning electron microscopy (SEM, FEI, Hillsboro, OR, USA). A volume of 2 mL 2 × 10^6^ CFU/mL *S. mutans* dilution or 2 × 10^4^ cell/mL *C. albicans* dilution was cultured to format single-species biofilm in a 24-well plate; and for dual-species biofilm, 2 × 10^6^ CFU/mL of *S. mutans* and 2 × 10^4^ cell/mL of *C. albicans* were used to form biofilms [3]. Each well of the 24-well plate contained a round glass slide (diameter = 14 mm) with 2 mL bacteria or fungal dilution for biofilm formation [41]. After 4 h anaerobic incubation, the biofilms were fixed with glutaraldehyde at room temperature for 12 h, then serially dehydrated in ethanol and tertiary butyl alcohol and sputter-coated with gold [41]. The specimens were examined at 500× and 10,000× magnification. Representative pictures are shown.

### 4.5. Crystal Violet (CV) Staining

The biomass of the biofilms with or without drugs was evaluated via CV staining, following previous studies [11,42]. Then *S. mutans* were diluted to 2 × 10^6^ CFU/mL, and *C. albicans* were diluted to 2 × 10^4^ cell/mL, and 2 mL dilution was cultured to format biofilms in 24-well plates. For dual-species biofilm, inoculum for the experiment was adjusted to 2 × 10^6^ CFU/mL of *S. mutans* and 2 × 10^4^ cell/mL of *C. albicans* [3]. After incubation for 4 h, the plates were washed twice with PBS [11]. The biofilms were air-dried and then stained with 0.1% (*w*/*v*) CV for 5 min at room temperature, washed three times with PBS to remove the unbound stain, dried and dissolved in 1 mL of 33% acetic acid [11]. The biofilm biomass was quantified by the optical density measured at a wavelength of 600 nm (OD_600nm_) [41]. The experiments were performed for six times in each group.

### 4.6. Polysaccharide Synthesis

The phenol-sulfuric acid method was used to measure the water-insoluble polysaccharides of biofilms [15]. Overnight cultures of *S. mutans* and *C. albicans* were respectively adjusted to 2 × 10^6^ CFU/mL and 2 × 10^4^ cell/mL for single- or dual-species biofilm formation. Biofilms with or without drugs were collected by scraping, and then, they were centrifuged (12,000 rpm) for 5 min at 4 °C and washed twice with PBS [15]. The precipitate was then resuspended in 1 M NaOH solution, placed in a constant temperature water bath at 37 °C and incubated for 3 h. Volumes of 100 μL of 6% phenol solution and 0.5 mL of 95–97% sulfuric acid were added, followed by incubation for 30 min [11]. Then, 200 μL of the solution was transferred into a 96-well plate, and OD_620nm_ was determined with the microplate reader (SpectraMax M5, Molecular Devices, San Jose, CA, USA) [11,42]. Six glucose concentrations of 0, 10, 20, 30, 40 and 50 μg/mL were used to plot the standard curve of OD_620nm_ readings to polysaccharide concentrations.

### 4.7. Lactic Acid Secretion

Overnight, bacterial cultures of *S. mutans* were diluted to 2 × 10^6^ CFU/mL into fresh BHIS for biofilm formation with or without drugs. A volume of 2 mL 2 × 10^4^ cell/mL *C. albicans* dilution was used for biofilm formation. For dual-species biofilm, inoculum for the experiment was adjusted to 2 × 10^6^ CFU/mL of *S. mutans* and 2 × 10^4^ cell/mL of *C. albicans* [3]. After incubation, the biofilms in 24-well plates were rinsed twice with PBS and then immersed in 1.5 mL buffered peptone water (BPW, Sigma-Aldrich, Saint Louis, MO, USA) with 0.2% sucrose and incubated at 37 °C for 3 h (*n* = 6) [15,42]. After removing planktonic cells by centrifugation, the supernatants were decanted to measure lactate concentrations according to the manuscript of the Lactate Assay Kit (MAK064, Sigma-Aldrich, Saint Louis, MO, USA) [43]. The absorbance at 530 nm was recorded using a microplate spectrophotometer, and lactate concentrations were calculated by the standard curves [15].

### 4.8. Biofilm Viability Using the MTT Assay

The 3-(4,5-dimethylthiazol-2-yl)-2,5-diphenyl tetrazolium bromide (MTT) (VWR Chemicals, OH, USA) assay was used to estimate the viability of bacteria and fungi in biofilms [15]. *S. mutans* were diluted to 2 × 10^6^ CFU/mL in BHIS and *C. albicans* were diluted to 2 × 10^4^ cell/mL in YNBB to form biofilm, with or without drugs. For dual-species biofilm, inoculum for the experiment was adjusted to 2 × 10^6^ CFU/mL of *S. mutans* and 2 × 10^4^ cell/mL of *C. albicans* [3]. Biofilms with or without drugs were washed twice with PBS (*n* = 6). A volume of 1mL MTT dye (0.5 mg/mL MTT in PBS) was added into each well and incubated at 37 °C under 5% CO_2_ for 1 h. A volume of 1 mL dimethyl sulfoxide (DMSO) was added in each well at room temperature for 20 min to dissolve the formazan crystals. After mixing via pipetting, 200 μL of the DMSO solution was collected and transferred into 96-well plate. OD_570nm_ was determined using the microplate reader (SpectraMax M5, Molecular Devices, Sunnyvale, CA, USA).

### 4.9. Biofilm CFU Counts

Six wells of each group in 24-well plates were used for CFU counting [15]. The *S. mutans* were diluted to 2 × 10^6^ CFU/mL into fresh BHIS, and 2 mL bacteria dilution was cultured to form biofilms in 24-well plates. A volume of 2 mL 2 × 10^6^ CFU/mL C. albicans dilution was cultured in YNBB. For dual-species biofilm, inoculum for the experiment was adjusted to 2 × 10^6^ CFU/mL of S. mutans, and 2 × 10^4^ cell/mL of *C. albicans* [3]. Biofilms with or without drugs were transferred into tubes with 2 mL of PBS, and the biofilms were harvested by scraping and sonication/vortexing (Fisher, Pittsburg, PA, USA) [15]. The suspensions were serially diluted and spread onto BHI or YPD agar plates. For dual-species, we added 8 μg/mL amphotericin B in BHI agar plates and 8 μg/mL gentamicin in YPD agar plates to inhibit the growth of *C. albicans* and *S. mutans*, respectively [44]. After a 2-day incubation at 37 °C under 5% CO_2_, the colony number was counted and then CFU counts were determined.

### 4.10. Confocal Laser Scanning Microscopy (CLSM)

For analyzing EPS production and distribution within the biofilms with or without drugs, we used CLSM for EPS/bacterial staining. *S. mutans* were diluted to 2 × 10^6^ CFU/mL into BHIS, and *C. albicans* were diluted to 2 × 10^4^ cell/mL into YNBB for biofilm formation. For dual-species biofilm, inoculum for the experiment was adjusted to 2 × 10^6^ CFU/mL of S. mutans, and 2 × 10^4^ cell/mL of *C. albicans* [3]. As in previous studies, 1 μM Alexa Fluor 647 (Invitrogen, Eugene, OR, USA) and 2.5 μM SYTO 9 (Invitrogen, Carlsbad, CA, USA) were used to label EPS and bacterial cells, respectively [41,45]. Biofilms were grown on coverslips and observed by CLSM (Olympus FV1000, Tokyo, Japan) at a range of 495–515 nm for SYTO 9 and 655–690 nm for Alexa Fluor 647 [41,45]. Images of six random fields of each group were captured. A three-dimensional reconstruction of the biofilms was analyzed, and the EPS/bacteria ratio was calculated. 

### 4.11. Quantitative Real-Time-Polymerase-Chain Reaction (qRT-PCR)

For qRT-PCR, UA159 biofilms were grown at 37 °C under anaerobic conditions (90% N_2_, 5% CO_2_, 5% H_2_) to 2 × 10^6^ CFU/mL in fresh BHIS with or without drugs. Dual-species biofilms were cultured adjusted to 2 × 10^6^ CFU/mL of *S. mutans* and 2 × 10^4^ cell/mL of *C. albicans* [3]. Biofilms were grown in 24-well plates for 4 h. qRT-PCR was used to quantify expression of selected genes in biofilms, with *gyrA* of *S. mutans* as an internal control [41]. The cells were harvested from the biofilms and snap frozen in liquid nitrogen until they were needed [15]. Total bacterial RNA isolation, purification and reverse transcription of complementary DNA (cDNA) were performed as previously described [15]. All primers for qRT-PCR were obtained commercially (Sangon Biotech, Shanghai, China) and are listed in Table 1. Threshold cycle values (CT) were determined, and the data were analyzed by BIO-RAD CFX MANAGER software (version 2.0, Hercules, CA, USA) using the 2^−ΔΔCT^ method [41].

### 4.12. Enamel Hardness Measurement

Extracted caries-free human teeth were obtained from the *** following a protocol approved by the *** Institutional Review Board. Teeth were cleaned and stored in 0.1% thymol solution at 4 °C before use. As previous study, enamel slabs were prepared to have a diameter of 6 mm and a thickness of 2 mm [42]. Except the enamel surfaces, the rest area was coated with two layers of acid-resistant nail varnish. The specimens were polished using sandpapers with grit of # 600, 1200, 2400 and 4000, consecutively, with copious water [42]. The 30 enamel slabs were randomly divided into five groups of 6 slabs each and incubated with or without drugs.

A hardness tester (HVS-10Z, Jingbo Company, Zhejiang, China) was used with a Vickers indenter, under a 50 g load with a dwell time of 20 s [42]. The area selected for indentation was the center of enamel surface. Every enamel specimen had six indentations. Measurements were conducted before and after biofilm acid attacks. The sterile specimens were placed in 24-well plates containing 2 mL 2 × 10^6^ CFU/mL of *S. mutans* and 2 × 10^4^ cell/mL *C. albicans* suspension, as a previous study described [42]. Each day, the slabs were placed into a well of 24-well plates and immersed in 2 mL of YNBB at pH 7.4 at 37 °C for 4 h. Then, they were removed and placed into a new plate with the YNBB-R medium which reduced the sucrose content at pH 7 for 20 h at 37 °C in 5% CO_2_. The daily medium change was done under aseptic conditions. Since this biofilm model took 14 days to finish, to avoid the biofilm becoming old and too thick which would deprive the interior bacteria from nutrients, a sterile paper was used to remove the biofilm from the slab every 24 h, which was then inoculated again to grow a new biofilm on the slab. This biofilm model and cyclic immersion treatment was repeated for 14 days.

### 4.13. Statistical Analysis

Data analyses were performed using Statistical Package for the Social Sciences (SPSS 22.0, Chicago, IL, USA). All data were expressed as the mean value ± standard deviation (mean ± sd). Statistical significance was analyzed by using the one-way analyses of variance (ANOVA) and Student–Newman–Keuls test. A confidence level of 95% (*p* < 0.05) was considered significant.

## 5. Conclusions

This study demonstrated for the first time that flavonoid baicalein had strong biofilm-suppression, caries-inhibition and enamel-protection capabilities. Baicalein showed the most potent and the greatest reduction in biofilm biomass, polysaccharide and lactic acid production among the test groups of baicalein, naringenin and catechin, and CHX. Baicalein reduced biofilm CFU by more than 4 orders of magnitude, and reduced biofilm acids by 99.99%, for *S. mutans*-*C. albicans* biofilm, compared to the UC group. In addition, the enamel hardness in the baicalein group showed only a minimal decrease under 14-day biofilm acid attacks, resulting in an enamel hardness that was 2.75 times greater than that of the untreated control group. Baicalein-based drugs are promising for a wide range of applications to suppress biofilms, prevent dental caries and protect tooth structures. 

## Figures and Tables

**Figure 1 ijms-23-10593-f001:**
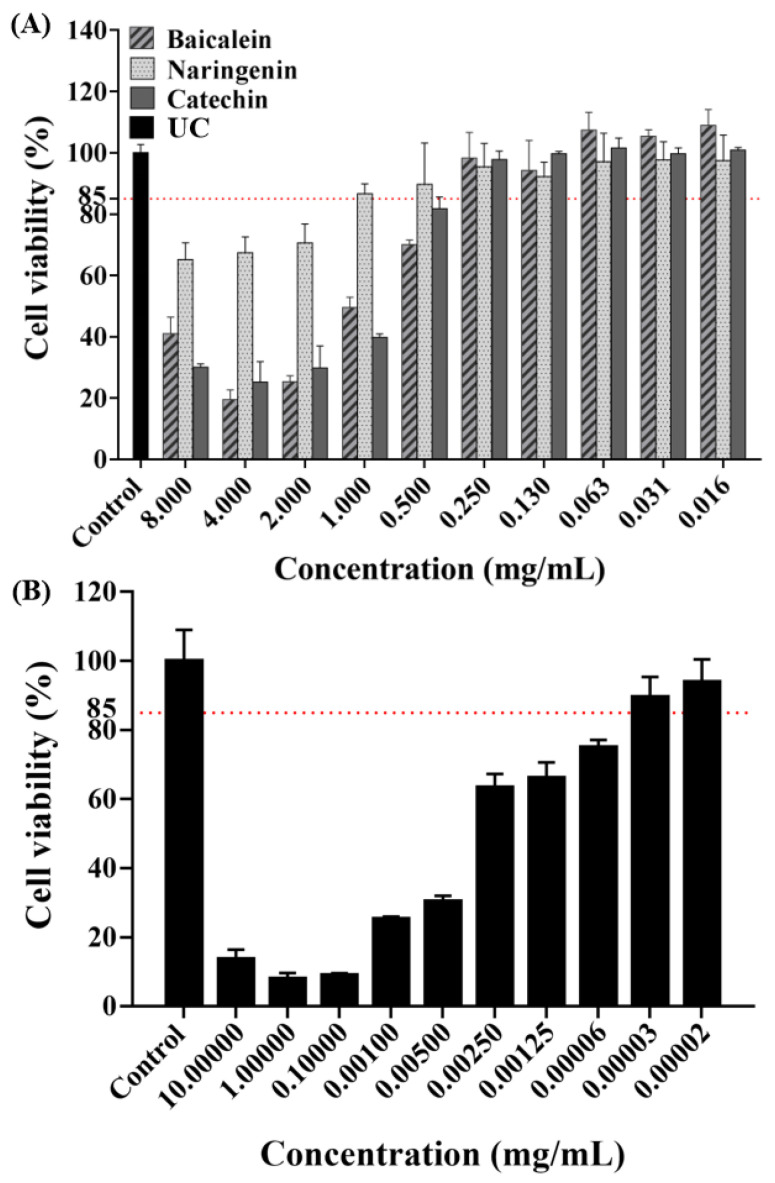
Cell viability tests of HOK cells after being exposed to flavonoids (**A**) or CHX; (**B**) (mean ± sd; *n* = 3).

**Figure 2 ijms-23-10593-f002:**
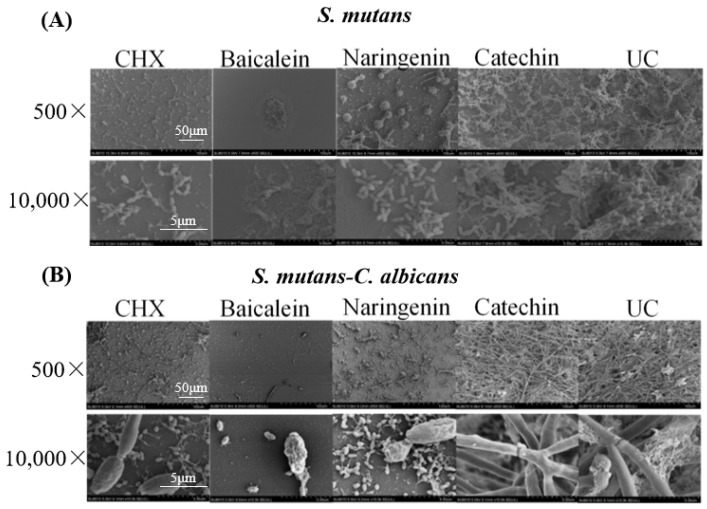
SEM observation of (**A**) *S. mutans* and (**B**) *S. mutans-C. albicans* biofilms after flavonoids or CHX treatment.

**Figure 3 ijms-23-10593-f003:**
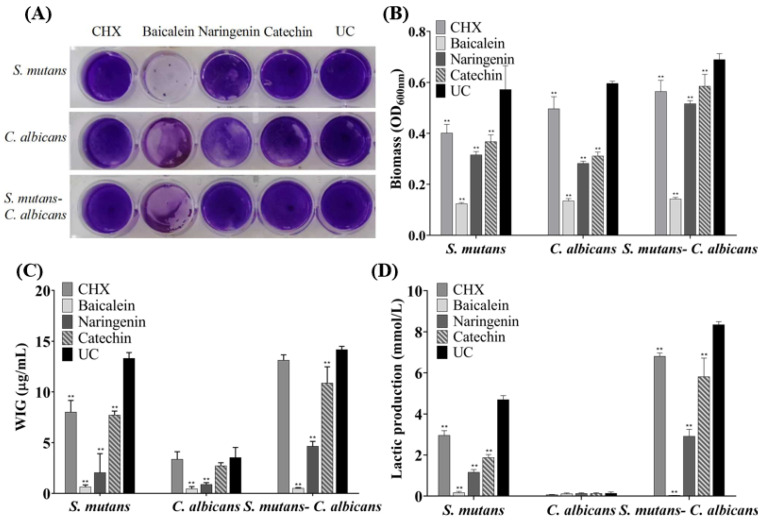
Biofilm formation with or without drugs: (**A**) a CV staining photograph of biofilms grown in a 24-well polystyrene microtiter plate; (**B**) quantification of biofilm biomass; (**C**) polysaccharide synthesis and (**D**) lactic acid production of the single-species and dual-species biofilms after flavonoids or CHX treatment. The values represent the mean ± sd for six independent experiments (mean ± sd, *n* = 6; ** *p* < 0.01).

**Figure 4 ijms-23-10593-f004:**
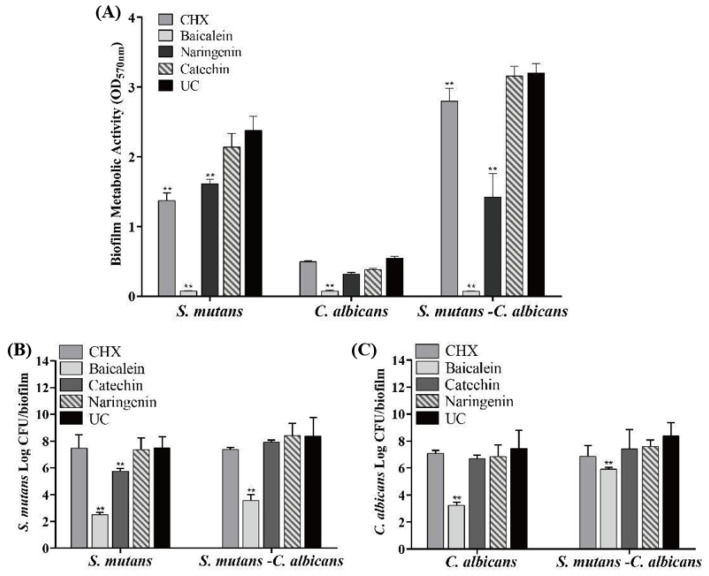
Antimicrobial effects of flavonoids or CHX (mean ± sd; *n* = 6; ** *p* < 0.01): (**A**) MTT metabolic activity, (**B**) *S. mutans* colony-forming units (CFU) in *S. mutans* and *S. mutans*-*C. albicans* biofilms, and (**C**) *C. albicans* CFU in *C. albicans* and *S. mutans*-*C. albicans* biofilms.

**Figure 5 ijms-23-10593-f005:**
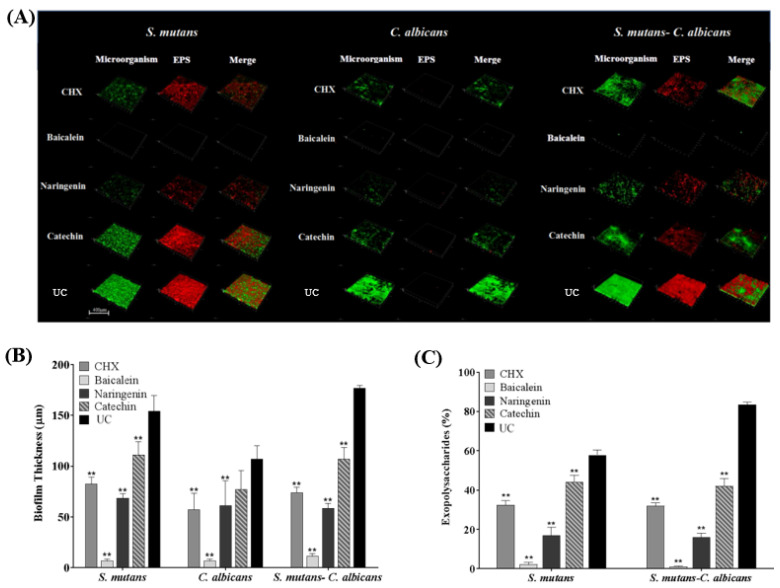
Representative CLSM staining images (**A**). Biofilm thickness (**B**) and EPS/bacterial or fungal ratio were calculated (**C**) (mean ± sd; *n* = 6; ** *p* < 0.01).

**Figure 6 ijms-23-10593-f006:**
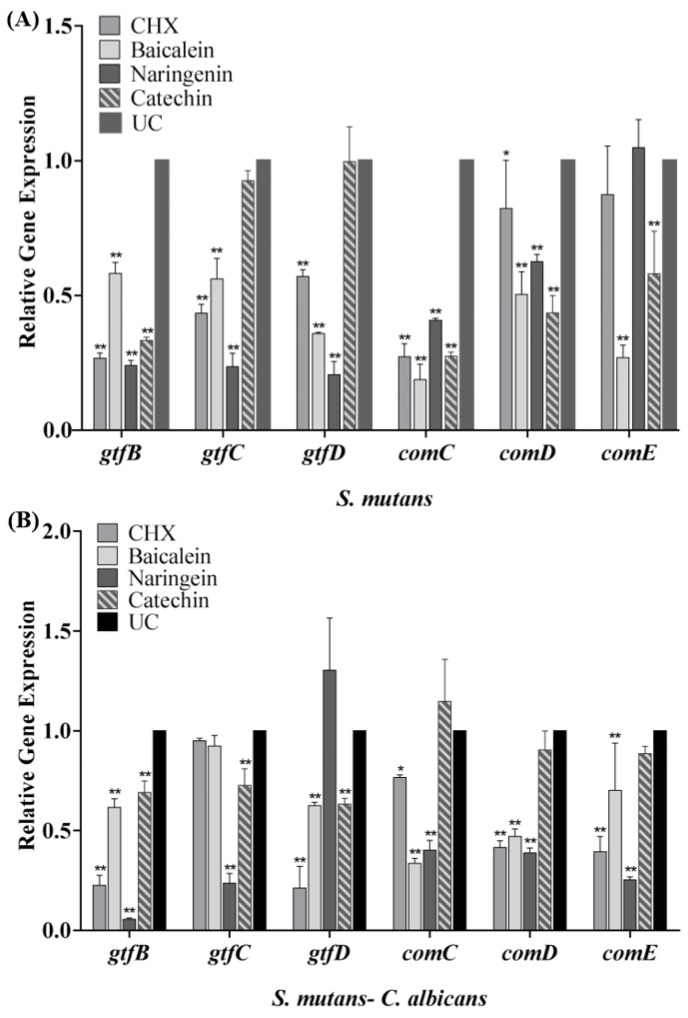
The *S. mutans gtfB*/*C*/*D*, *comC*/*D*/*E* gene expression in single-species (**A**) and dual-species biofilms (**B**) via qRT-PCR (mean ± sd, *n* = 3; * *p* < 0.05, ** *p* < 0.01).

**Figure 7 ijms-23-10593-f007:**
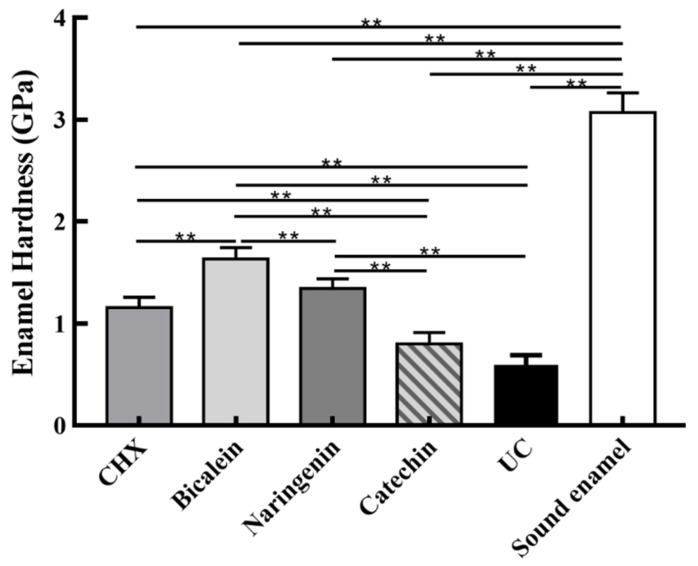
Enamel hardness at the enamel surface after *S. mutans*-*C. albicans* biofilm acid attack for 14 days (mean ± sd, *n* = 6; ** *p* < 0.01).

**Table 1 ijms-23-10593-t001:** Primers and probes used in qRT-PCR.

Primer	Nucleotide Sequence	Reference
*gyrA*-F	5′ ATTGTTGCTCGGGCTCTTCCAG 3′	[46]
*gyrA*-R	5′ ATGCGGCTTGTCAGGAGTAACC 3′	
*gtfB*-F	5′ ACACTTTCGGGTGGCTTG 3′	[47]
*gtfB*-R	5′ GCTTAGATGTCACTTCGGTTG 3′	
*gtfC*-F	5′ CCAAAATGGTATTATGGCTGTCG 3′	[47]
*gtfC*-R	5′ TGAGTCTCTATCAAAGTAACGCAG 3′	
*gtfD*-F	5′ AATGAAATTCGCAGCGGACTTGAG 3′	[48]
*gtfD*-R	5′ TTAGCCTGACGCATGTCTTCATTGTA 3′	
*comC*-F	5′ GACTTTAAAGAAATTAAGACTG 3′	[47]
*comC*-R	5′ AAGCTTGTGTAAAACTTCTGT 3′	
*comD*-F	5′ CTCTGATTGACCATTCTTCTGG 3′	[47]
*comD*-R	5′ CATTCTGAGTTTATGCCCCTC 3′	
*comE*-F	5′ CCTGAAAAGGGCAATCACCAG 3′	[47]
*comE*-R	5′ GGGGCATAAACTCAGAATGTGTCG 3′	

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
