# Peer review of "Flavonoid Baicalein Suppresses Oral Biofilms and Protects Enamel Hardness to Combat Dental Caries"

_ijms, 2022, doi:10.3390/ijms231810593_

Round 1

Reviewer 1 Report

It is a very interesting paper that should be published after addressing some minor concerns:

1. The cell count should be provided for each experiment.

2. The units of each agent should be same in Figure 1

3. English should be checked for a few of awkward places such as "And lactic acid production of baicalein on biofilms was relatively low.", "MTT metabolic activity 126 and colony forming unit (CFU) were markedly decreased in baicalein (p < 0.01).", "catechin showed significant no-cytotoxic effects on HOK cells", "the lactic acid secretion by biofilms 216 decreased significantly in drugs", etc. 

Author Response

Dear Reviewer:

I hope you are doing very well. Thank you very much for a favorable review on our paper, ijms-1902772, entitled: “Flavonoid baicalein suppresses oral biofilms and protects enamel hardness to combat dental caries.” The reviewers provided excellent comments, all of which have been addressed in the revised paper. Below is a point-by-point response to the review comments. Each comment is followed by our response in a red color (Please see the attachment). The revised sections in the paper are also highlighted in red.

Response to Reviewer Comments:

It is a very interesting paper that should be published after addressing some minor concerns:

Thank you very much for an excellent review that has helped significantly improve our paper.

  1. The cell count should be provided for each experiment.

Excellent point.

To address your comments, we had added following sentences in Section 4.2 of the Materials and Methods: " Then S. mutans were diluted to 2 × 106 CFU/mL into fresh BHIS, and C. albicans were diluted to 2 × 104 cell/mL into YNBB medium, with or without drugs.”

We had added some sentences in Section 4.4 of the Materials and Methods: "2mL 2 × 106 CFU/mL S. mutans dilution or 2 × 104 cell/mL C. albicans dilution were cultured to format single-species biofilm in 24-well plate; and for dual-species biofilm, 2 × 106 CFU/ml of S. mutans and 2 × 104 cell/mL of C. albicans were used to form biofilms [3].”

We had added some sentences in Section 4.5 of the Materials and Methods: " Then S. mutans were diluted to 2 × 106 CFU/mL, and C. albicans were diluted to 2 × 104 cell/mL, and 2 mL dilution were cultured to format biofilms in 24-well plates. For dual-species biofilm, inoculum for the experiment was adjusted to 2 × 106 CFU/ml of S. mutans, and 2 × 104 cell/mL of C. albicans [3].”

We had added some sentences in Section 4.6 of the Materials and Methods: " Overnight cultures of S. mutans and C. albicans were respective adjusted to 2 × 106 CFU/mL and 2 × 104 cell/mL for single- or dual-species biofilm formation.”

We had added some sentences in Section 4.7 of the Materials and Methods: " Overnight bacterial cultures of S. mutans were diluted to 2 × 106 CFU/mL into fresh BHIS for biofilm formation with or without drugs. 2 mL 2 × 104 cell/mL C. albicans dilution were used for biofilm formation. For dual-species biofilm, inoculum for the experiment was adjusted to 2 × 106 CFU/ml of S. mutans, and 2 × 104 cell/mL of C. albicans [3].”

We had added some sentences in Section 4.8 of the Materials and Methods: " S. mutans were diluted to 2 × 106 CFU/mL in BHIS and C. albicans were diluted to 2 × 104 cell/mL in YNBB to form biofilm, with or without drugs. For dual-species biofilm, inoculum for the experiment was adjusted to 2 × 106 CFU/ml of S. mutans, and 2 × 104 cell/mL of C. albicans [3].”

We had added some sentences in Section 4.9 of the Materials and Methods: " The S. mutans were diluted to 2 × 106 CFU/mL into fresh BHIS and 2 mL bacteria dilution were cultured to form biofilms in 24-well plates. 2 mL 2 × 106 CFU/mL C. albicans dilution were cultured in YNBB. For dual-species biofilm, inoculum for the experiment was adjusted to 2 × 106 CFU/ml of S. mutans, and 2 × 104 cell/mL of C. albicans [3].”

We had added some sentences in Section 4.10 of the Materials and Methods: " S. mutans were diluted to 2 × 106 CFU/mL into BHIS and C. albicans were diluted to 2 × 104 cell/mL into YNBB for biofilm formation. For dual-species biofilm, inoculum for the experiment was adjusted to 2 × 106 CFU/ml of S. mutans, and 2 × 104 cell/mL of C. albicans [3].”

We had added some sentences in Section 4.11 of the Materials and Methods: " For qRT-PCR, UA159 biofilms were grown at 37°C under anaerobic conditions (90% N2, 5% CO2, 5% H2) to 2 × 106 CFU/mL in fresh BHIS with or without drugs. Dual-species biofilms were cultured adjusted to 2 × 106 CFU/ml of S. mutans and 2 × 104 cell/mL of C. albicans [3]. Biofilms were grown in 24-well plates for 4 h.”

We had added some sentences in 2nd Paragraph of Section 4.12 of the Materials and Methods: " The sterile specimens were placed in 24-well plates containing 2 mL 2 × 106 CFU/ml of S. mutans and 2 × 104 cell/mL C. albicans suspension, as previously study described [42].”

  1. The units of each agent should be same in Figure 1

Excellent point. To address your comments, we have revised Fig. 1 as following:

And we have changed the sentence to: “Based on the cell cytotoxicity test, the antibacterial concentrations were selected for our study as follows: 0.250 mg/mL of baicalein, 1.000 mg/mL of naringenin, 0.250 mg/mL of catechin, and 0.00031 mg/mL of CHX, respectively.” In 1st paragraph of the Results.

  1. English should be checked for a few of awkward places such as "And lactic acid production of baicalein on biofilms was relativelylow.", "MTT metabolic activity 126 and colony forming unit (CFU) were markedly decreased in baicalein (p < 0.01).", "catechin showed significant no-cytotoxic effects on HOK cells", "the lactic acid secretion by biofilms 216 decreased significantly in drugs", etc. 

Thank you very much for an excellent review, and we have revised and improved language throughout the article.

We revised some sentences as following:

In 5th paragraph of the Results: “And lactic acid production of baicalein group on biofilms was lowest in all groups.”

In 6th paragraph of the Results: “MTT metabolic activity and colony forming unit (CFU) were markedly decreased in baicalein group (p < 0.01).” 

In 2nd paragraph of the Discussion: “And results indicated that flavonoids concentrations of 0.250 mg/mL in baicalein, 1.000 mg/mL in naringenin, and 0.250 mg/mL in catechin showed good biocompatibility on HOK cells.”

In 5th paragraph of the Discussion: “In the present study, the lactic acid secretion by biofilms decreased significantly in all drug groups, especially baicalein group.”

We sincerely thank you for the excellent comments and advices.

Thank you very much for your consideration. We look forward to hearing from you.

Best Regards,

Hockin Xu

Professor, Director, Biomaterials & Tissue Engineering Division

Reviewer 2 Report

Please see the pdf file where I have included all my comments and highlights in color.

Author Response

September 2, 2022

Huakun (Hockin) Xu

Professor, Director, Biomaterials & Tissue Engineering Division

Department of Advanced Oral Sciences and Therapeutics

University of Maryland Dental School, Baltimore, MD 21201, USA

Email:  hxu@umaryland.edu   

Dear Reviewer:

I hope you are doing very well. Thank you very much for a favorable review on our paper, ijms-1902772, entitled: “Flavonoid baicalein suppresses oral biofilms and protects enamel hardness to combat dental caries.” The reviewers provided excellent comments, all of which have been addressed in the revised paper. Below is a point-by-point response to the review comments. Each comment is followed by our response in a red color. The revised sections in the paper are also highlighted in red.

Response to Reviewer Comments:

Thank you very much for an excellent review that has helped significantly improve our paper.

Introduction

-this is not true. Here is at least one paper:

https://doi.org/10.1016/j.micpath.2017.03.033

Excellent point. As your suggestion, we have changed the sentence to: “In addition, to date, there were few studies on the role of flavonoids in affecting S. mutans, C. albicans and the dual-species biofilm formation [21, 22].” in 3rd paragraph of the Introduction.

Why would you even start an experiment if you suspected it was the right thing to do in the first place?

Excellent point. Thanks for your reminder, we have deleted the sentence: “The null hypothesis was that flavonoids would have no influence on biofilm formation of S. mutans and C. albicans.” in 4th paragraph of the Introduction.

Results

- There is an error here. Diagram c is actually d, and d is c.

Excellent point. Thanks for your reminder, we have revised the Fig. 3 legend to: “Figure 3. Biofilm formation with or without drugs: (A) A CV staining photograph of biofilms grown in a 24-well polystyrene microtiter plate; (B) Quantification of biofilm biomass; (C) Polysaccharide synthesis and (D) Lactic acid production of the single-species and dual-species biofilms after flavonoids or CHX treatment. The values represent the mean±sd for six independent experiments (mean±sd, n= 6; **p < 0.01).”

- whether it is necessary to specify p < 0.05 if you have it nowhere?

Excellent point. As you suggested, we have revised the last sentence of Fig. 3 legend to: “The values represent the mean±sd for six independent experiments (mean±sd, n= 6; **p < 0.01).” 

- The sentence is not clear enough. Do you mean....

Compared with the UC group, the CFU of the S. mutans biofilm decreased by 5 logs and 4 logs in the S. mutans and S. mutans-C. albicans biofilms, respectively, at 0.250 mg/ml baicalein (Fig. 4B). While 4 logs and 1 log in C. albican and S. mutans-C. albicans biofilms were reduced for C. albicans CFU at 0.250mg/mL baicalein (Fig. 4C). The use of baicalein caused the greatest reduction in biofilm activity, while the CHX group appeared to have no effect.

Excellent point. As you suggested, we have revised 6th Paragraph in the Results to: “Compared with the UC group, the CFU of the S. mutans biofilm was decreased by nearly 5 logs and 4 logs in S. mutans and S. mutans-C. albicans biofilms, respectively, at 0.250mg/mL baicalein (Fig. 4B). While 4 logs and 1 log in C. albicans and S. mutans-C. albicans biofilms were reduced for C. albicans CFU at 0.250mg/mL baicalein (Fig. 4C). The use of baicalein caused the greatest reduction in biofilm activity, while the CHX group appeared to have no effect.”

- How come you get different results for the same sample group (S.mutans-C.albicans) (diagram B and C)?

Excellent point. For dual-species biofilms, we used BHI agar plates with 8 μg/mL amphotericin B and YPD agar plates with 8 μg/mL gentamicin to obtain the CFU of S. mutans and C. albicans, respectively [43]. Therefore, we presented the S. mutans CFU of S.mutans-C.albicans in diagram B and the C. albicans CFU of S.mutans-C.albicans in diagram C.

-Please, add the name for second sample group

Excellent point. To addressed your comments, we have revised Fig. 4 legend to: “Figure 4. Antimicrobial effects of flavonoids or CHX (mean±sd; n = 6; **p < 0.01): (A) MTT metabolic activity, (B) S. mutans colony-forming units (CFU) in S. mutans and S. mutans-C. albicans biofilms, and (C) C. albicans CFU in C. albicans and S. mutans-C. albicans biofilms.”

-How come there are no results for C. albicans in diagram c?

Excellent point. As shown in Fig. 5A, there is almost no EPS production in all groups, therefore we have deleted the results for C. albicans in diagram C of Fig. 5. And we add the sentence in the 7th Paragraph of Results: “And there was almost no EPS production of C. albicans biofilms in all groups, therefore we have deleted the results for C. albicans biofilms in Fig. 5C.”

-Is this correct? Enamel hardness has tripled? Can you back this up with exact numbers, since it is not apparent from the chart?

Excellent point. Thanks for your reminder, we have changed the sentence to: “These results showed that: (1) Flavonoids baicalein substantially increased the enamel hardness, by 2.75 folds, compared to the UC group; (2) The efficacy of baicalein and naringenin was much better than CHX in protecting the enamel hardness.” in 9th paragraph of the Results.

Discussion

- The cited literature (25) does not correspond to this content.

Excellent point. Thanks for your reminder, we have revised the sentences as following: “According to present studies [23, 27], we obtained the drug concentrations of more than 85% cell viability. And results indicated that flavonoids concentrations of 0.250 mg/mL in baicalein, 1.000 mg/mL in naringenin, and 0.250 mg/mL in catechin showed good biocompatibility on HOK cells. Therefore, the concentrations above were used in the following experiments.” in 2nd paragraph of the Discussion.

-Almost the same sentence as in the results; repetition of saideog.

Excellent point. Thanks for your reminder, we have changed these sentences to: “In the present study, flavonoids baicalein substantially reduced the biofilm metabolic activity, by more than 4 orders of magnitude. In sharp contrast, the biofilm CFU reduction in CHX had no significance compared to the UC group. The S. mutans and C. albicans in biofilms may be resistant to CHX [11]. Therefore, baicalein possesses the anti-microbe effects of flavonoids are much better than CHX.” in 7th Paragraph of the Discussion.

-I think this is a bit "too strong" a conclusion based on a one type of analysis.

Excellent point. Thanks for your reminder, we have changed the sentence to: “In other words, naringenin and baicalein may inhibit the synthesis of QS system, which would reduce the internal signal transduction in S. mutans or S. mutans and C. albicans.” in 7th Paragraph of the Discussion.

Materials and Methods

-Different line space

Excellent point. Thanks for your reminder, we have modified the line space of 4.4. Section in the Materials and Methods as other sections.

-As in previous atudy………

Excellent point. As you suggested, we have improved the following sentences in 4.10. Section of the Materials and Methods: “As in previous studies, 1 μM Alexa Fluor 647 (Invitrogen, Eugene, OR, USA) and 2.5 μM SYTO 9 (Invitrogen, Carlsbad, CA, USA) were used to label EPS and bacterial cells, respectively [38, 42].”

- Where is Table 1.?

Excellent point. To address your comments, we had added Table 1 in 4.11. Section of the Materials and Methods:

Table 1. Primers and probes used in qRT-PCR

Primer

Nucleotide sequence

Reference

gyrA-F

5′ ATTGTTGCTCGGGCTCTTCCAG 3′

[43]

gyrA-R

5′ ATGCGGCTTGTCAGGAGTAACC 3′

gtfB-F

5′ ACACTTTCGGGTGGCTTG 3′ 

[44]

gtfB-R

5′ GCTTAGATGTCACTTCGGTTG 3′

gtfC-F

5′ CCAAAATGGTATTATGGCTGTCG 3′

[44]

gtfC-R

5′ TGAGTCTCTATCAAAGTAACGCAG 3′

gtfD-F

5′ AATGAAATTCGCAGCGGACTTGAG 3′

[45]

gtfD-R

5′ TTAGCCTGACGCATGTCTTCATTGTA 3′

comC-F

5′ GACTTTAAAGAAATTAAGACTG 3′

[44]

comC-R

5′ AAGCTTGTGTAAAACTTCTGT 3′

comD-F

5′ CTCTGATTGACCATTCTTCTGG 3′

[44]

comD-R

5′ CATTCTGAGTTTATGCCCCTC 3′

comE-F

5′ CCTGAAAAGGGCAATCACCAG 3′

[44]

comE-R

5′ GGGGCATAAACTCAGAATGTGTCG 3′

- ? Please, explain this.

This is not correct neither used in this experiment.

Excellent point. As your suggestion, we have changed the sentence to: “Statistical significance was analyzed by using the one-way analyses of variance (ANOVA) and Student-Newman-Keuls test. A confidence level of 95% (p < 0.05) was considered significant. ” in 4.13. Section of the Materials and Methods.

Thank you very much for an insightful review and the constructive comments that have helped significantly improve our paper.

Thank you very much for your consideration. We look forward to hearing from you.

Best Regards,

Hockin Xu

Professor, Director, Biomaterials & Tissue Engineering Division
